# Dose- and Sex-Dependent Bidirectional Relationship between Intravenous Fentanyl Self-Administration and Gut Microbiota

**DOI:** 10.3390/microorganisms10061127

**Published:** 2022-05-30

**Authors:** Michelle Ren, Shahrdad Lotfipour

**Affiliations:** 1Department of Pharmaceutical Sciences, School of Pharmacy and Pharmaceutical Sciences, University of California Irvine, Irvine, CA 92697, USA; shahrdad@hs.uci.edu; 2Department of Emergency Medicine, School of Medicine, University of California Irvine, Irvine, CA 92697, USA; 3Department of Pathology and Laboratory Medicine, School of Medicine, University of California Irvine, Irvine, CA 92697, USA

**Keywords:** addiction, reward, reinforcement, motivation, bacteria, opioids, rats, gut–brain axis

## Abstract

Gut bacteria influence neural circuits in addiction-related behaviors. Given the association between opioid use, gastrointestinal distress, and microbial dysbiosis in humans and mice, we test the hypothesis that interactions between gut bacteria and the brain mediate the rewarding and reinforcing properties of fentanyl. We implant rats with intravenous catheters in preparation for fentanyl intravenous self-administration (IVSA) on an escalating schedule of reinforcement to determine factors that influence fentanyl intake, including sex, dose, and gut microbiota. Our data show the impact of fentanyl IVSA on gut microbiota diversity, as well as the role of gut microbiota on fentanyl IVSA, in Sprague Dawley rats in a sex- and dose-dependent manner (*n* = 10–16/group). We found that the diversity of gut microbiota within females dose-dependently predicts progressive but not fixed ratio schedules of fentanyl IVSA. Depending on sex and fentanyl dose, alpha diversity (richness and evenness measured with Shannon index) is either increased or decreased following fentanyl IVSA and predicts progressive ratio breakpoint. Our findings collectively suggest a role of gut bacteria in drug-related behavior, including motivation and reinforcement. This work provides feasibility for an intravenous fentanyl self-administration model and uncovers potential factors mediating drug use, which may lead to the development of effective addiction interventions.

## 1. Introduction

The United States is currently in an opioid crisis: the number of drug-related deaths has been rapidly increasing every year, with most cases involving fentanyl [1]. It is necessary to investigate mechanisms mediating the abuse potential of fentanyl to contribute to the development of successful treatment strategies. In this study, we evaluate the role of gut microbiota as a potential mechanism in fentanyl reward and reinforcement. A link between gut health and opioid intake is evident in the ability of opioids to significantly impact gastrointestinal function, (i.e., opioid-induced constipation). Indeed, opioid use is associated with imbalances in gut bacteria in both humans [2,3,4,5,6,7] and animals [8,9,10,11,12,13]. Preclinical studies have also shown an important role of gut microbiota in opioid-dependent behaviors, including reward, tolerance, and withdrawal [11,12,14].

The gut and brain communicate via the trillions of bacteria that reside in the gastrointestinal tract, collectively known as the gut microbiome. A growing field of literature supports a bidirectional relationship between gut bacteria and host health, particularly related to emotions, behavior, and neuropsychiatric disorders [15,16,17]. Gut microbiota provides a protective role by occupying intestinal surfaces and preventing overgrowth of pathogenic microorganisms [18]. An unbalanced gut environment, or gut dysbiosis, is bidirectionally associated with depression, anxiety, and learning and memory deficits [19,20,21]. Evidence also suggests a role of the microbiota–gut–brain axis in reward and motivation [22,23,24].

The present study aims to explore gut–brain interactions in fentanyl use, which has never been studied before in the literature. Because the literature is limited on fentanyl self-administration, one primary objective of this study is to establish a dose–response relationship of fentanyl intravenous self-administration (IVSA) in adult wild-type Sprague Dawley rats. We implant rats with intravenous catheters in preparation for fentanyl IVSA on an escalating schedule of reinforcement to determine factors that influence fentanyl intake, including sex and dose. Moreover, we test the hypothesis that fentanyl IVSA decreases gut bacteria’s alpha and beta diversity based on prior findings that chronic opioid treatment decreases both measures of diversity [12,13]. Given the bidirectional relationship between gut bacteria and behavior, we test the hypothesis that the diversity of gut bacteria predicts fentanyl IVSA. By identifying how gut bacteria influence fentanyl use, our research has the potential to significantly progress our understanding of the mechanisms influencing opioid addiction.

## 2. Materials and Methods

Animals: Adult Sprague Dawley rats (8–9 weeks of age) were obtained from Charles River (San Diego, CA, USA) and acclimated to our vivarium at least 7 days prior to experimentation. A total of 52 animals were used in this study with 10 animals excluded from data due to technical issues or lack of catheter patency, and 1 exclusion determined as an outlier via box-and-whisker plots. The final dataset analyzed includes 23 males (*n* = 9, 8, and 6 at 0, 1.25, and 2.5 μg/kg/infusion, respectively), and 18 females (*n* = 6, 8, and 4 at 0, 1.25, and 2.5 μg/kg/infusion). Animals were pair-housed (2/cage, 26 cages in final data) in a humidity and temperature-controlled room, and the assigned drug dose for each animal was cage-matched, (i.e., all animals housed in the same cage self-administered the same drug dose). Food and water were available ad libitum until 24-h prior to drug self-administration and onward when food was restricted to 20–25 g/animal/day (sufficient to maintain adult weight) to motivate exploration. Animals were randomly assigned to experimental groups using a random sequence generator. Animals were housed on a 12-h light-dark cycle (lights on at 07:00) and experimentation was performed during the light part of the light-dark cycle. All animal procedures were approved by the Institutional Animal Care and Use Committee at the University of California Irvine and performed in accordance with the Association for Assessment and Accreditation of Laboratory Animal Care.

Catheter Implantation: All rats were anesthetized with Equithesin (0.35 mL/100 g, intraperitoneal) and catheters implanted into their right external jugular vein in preparation for intravenous (i.v.) fentanyl self-administration (implantation surgery described previously) [25]. Animals were given carprofen (4 mg/kg, subcutaneous) (Med-Vet, Mettawa, IL, USA), a postoperative analgesic, immediately after surgery. Catheters were flushed daily with sterile heparinized saline (0.6 mL of 1000 units/mL heparin in 30 mL saline) to maintain patency. Animals were allowed at least 2 days to recover from surgery before drug self-administration.

Fentanyl Intravenous Self-Administration: Aqueous fentanyl citrate (Patterson Veterinary, Greeley, CO, USA) was mixed with saline and filtered through a 0.22 μm sterile filter (VWR, Radnor, PA, USA) prior to use. All animals were allowed to self-administer fentanyl at 0, 1.25, or 2.5 μg/kg/infusion during daily 2-h sessions for 5 days at a fixed ratio (FR) 1 schedule of reinforcement, 2 days at FR2, followed by 2 days on FR5, all with a 20-s timeout. Subsequently, animals were assessed for drug reward using a progressive ratio (PR) schedule of reinforcement, where the response requirement to earn an injection escalates within the same session according to an approximately logarithmic series [26]. Optimal fentanyl doses were selected from prior rodent studies showing acquisition of self-administration while avoiding potentially toxic effects of long-session and long-term self-administration access [27,28,29]. Animals self-administered fentanyl in individual chambers by poking their nose into a “reinforced” nose poke hole, which signaled a cue light to turn on and delivery of one 20 μL i.v. infusion of the assigned dose of fentanyl. Non-reinforced nose pokes at a second hole were counted to control for non-specific drug effects. Data were collected by a multichannel computer system (Med Associates, St. Albans, VT, USA). After the final session on each schedule of reinforcement, propofol (0.1 mL) (Zoetis, Parsipanny, NJ, USA) was injected through the catheter to test patency as indicated by rapid (5–10 s) anesthesia. Data were discarded from animals not demonstrating rapid anesthesia.

16S Sequencing: Fecal samples were suspended in a 1.7 mL Eppendorf tube prefilled with DNA/RNA Shield (Zymo Research, Irvine, CA, USA) and stored at −80 °C until processing. Bacterial genomic DNA from all samples was isolated using the Zymobiomics DNA Mini Kit in a 96-well format (Zymo Research, Irvine, CA, USA). The gDNA was used to target the 16S rRNA gene. A 16S rRNA amplicon PCR was performed targeting the V4-V5 region using the EMP primers (515F (barcoded) and 926R) [30,31]. The library was sequenced at the University of California Irvine’s Genomics High Throughput Facility using a MiSeq v3 chemistry with a PE300 sequencing length. Sequencing resulted in 12M single-end reads (forward) passing filter of which 11% are PhiX with a >Q30 = 85%. The raw forward sequences were imported into QIIME2 [32].

Statistical Analysis: Behavioral data were analyzed with JMP (SAS Institute, Cary, NC, USA). Each self-administration schedule of reinforcement (FR1, FR2, FR5, PR) was analyzed separately with multivariate ANOVA on sex, fentanyl dose, and response, with repeated measures on response (reinforced vs. non-reinforced). Any main effects were further analyzed using Bonferroni-corrected paired (response) or unpaired (drug) *t*-test post hoc comparisons. One outlier determined from box-and-whisker plots on days 10–11 mean PR infusions was removed from the data. Bacterial sequence data were analyzed using QIIME2 and R Studio.

Experimental Timeline: Adult rats were implanted with an indwelling catheter in their right external jugular vein. Following 2–3 days of post-operative recovery, animals were allowed to self-administer fentanyl via reinforced nose pokes as described above. One fecal sample was collected from each animal the day before fentanyl self-administration and the last day of self-administration to assess the impact of fentanyl IVSA on gut microbiota diversity and composition (Figure 1).

## 3. Results

### 3.1. Intravenous Fentanyl Self-Administration Is Dependent on Fentanyl Dose

Current data on fentanyl intravenous self-administration (IVSA) are limited, so we investigated responding for fentanyl infusions on an escalating schedule of reinforcement in wild-type Sprague Dawley rats to establish a fentanyl dose-response relationship. We analyzed fentanyl IVSA with a three-way ANOVA to investigate the roles of sex and fentanyl dose in fentanyl self-administration, in addition to discrimination between reinforced and non-reinforced nose pokes. Data were analyzed separately by schedule of reinforcement, with the mean responses of the last 2 days of each schedule, due to stabilization of learning and drug acquisition. We found a main effect of fentanyl dose across all schedules of reinforcement (fixed ratio (FR) 1: F(5,35) = 1.69, *p* = 0.05; FR2: F(5,35) = 2.43, *p* = 0.01; FR5: F(5,35) = 2.26, *p* = 0.01). Data are collapsed by sex due to a lack of sex differences (FR1: *p* = 0.62, FR2 and FR5: *p* = 0.82). Rats self-administered significantly more fentanyl at 1.25 vs. 0 μg/kg/infusion at FR1 (*p* = 0.01), FR2 (*p* = 0.002), and FR5 (*p* = 0.003) (Figure 2A, Table 1).

### 3.2. Motivation for Fentanyl Infusions Is Highest at 1.25 vs. 0 or 2.5 ug/kg/Infusion

The “breakpoint” at a progressive ratio (PR) schedule of reinforcement is a measure at which an animal stops responding for a reinforcer. We define breakpoint as the last ratio of responses successfully completed by an animal [26]. As fixed ratio is a measure of reinforcement rather than reward or motivation, we measured breakpoint to determine the motivation of animals to self-administer fentanyl infusions. A two-way ANOVA analyzing sex and fentanyl dose on mean infusions over 2 days at PR yielded a main effect of fentanyl dose (F(5,35) = 5.08, *p* = 0.0001). Post-hoc analysis showed significantly higher infusions self-administered at 1.25 vs. 0 μg/kg/infusion (*p* < 0.0001) and 2.5 vs. 0 μg/kg/infusion (*p* = 0.008) (Figure 2B, Table 1). The data shown are collapsed by sex, as we found no significant effect of sex (*p* = 0.75).

### 3.3. Fentanyl Self-Administration Changes Gut Bacteria Alpha Diversity

We isolated bacterial DNA from fecal samples collected from animals before and after self-administration to evaluate whether fentanyl IVSA alters the diversity of gut bacteria. Alpha diversity was measured using two diversity indices: Chao1, which estimates the total richness of a microbial sample based on relative abundance, and Shannon, which considers both richness and evenness of a microbial sample. We observed a main effect of sex between subjects (F(1,35) = 6.88, *p* = 0.01), a sex * fentanyl dose interaction (F(2,35) = 3.52, *p* = 0.04), and a main effect of time within subjects (before vs. after fentanyl IVSA) (F(1,35) = 4.07, *p* = 0.05). When evaluating post hoc comparisons, there was a significant reduction of alpha diversity in males compared to females before fentanyl IVSA (Shannon: *p* = 0.02, Chao1: *p* = 0.01). This baseline difference in alpha diversity appears to be driven by Verrumicrobia (*p* = 0.01), Prevotella (*p* = 0.02), and Akkermansia (*p* = 0.01), as relative abundance in these bacterial groups was significantly different in females than males before fentanyl IVSA (Table 2). When analyzing samples by sex and fentanyl dose, we found that Shannon diversity (richness and evenness) is increased after vs. before IVSA in males at 1.25 μg/kg/infusion (*p* = 0.02), and Chao1 diversity (richness) is increased after IVSA at 0 and 1.25 μg/kg/infusion (*p* = 0.03 and *p* = 0.02, respectively) (Figure 3A,B). Further, we observed that fentanyl IVSA at 1.25 vs. 0 μg/kg/infusion reduced alpha diversity in females (*p* = 0.05). Firmicutes/Bacteroidetes ratios remained stable before and after fentanyl IVSA (Figure 3C). In males that self-administered fentanyl at 1.25 μg/kg/infusion, changes in alpha diversity were seen in the bacterial phylum Verrucomicrobia (*p* = 0.03) and genera Ruminococcus (*p* = 0.03) and Akkermansia (*p* = 0.03) (Table 2). Relative abundance of the bacterial genus Prevotella was greater in females that self-administered 1.25 μg/kg/infusion (Table 2). There were no significant differences in relative abundance or percent composition of Firmicutes, Bacteroidetes, Proteobacteria, Tenericutes, Actinobacteria, or Lactobacillus before and after fentanyl IVSA (data not shown).

### 3.4. Baseline Gut Bacteria Alpha Diversity Does Not Predict Fentanyl Self-Administration

To investigate whether an individual animal’s gut bacteria diversity impacts fentanyl self-administration behavior, we analyzed the correlation between gut bacteria alpha diversity before fentanyl IVSA and (1) mean infusions self-administered at FR1 and (2) mean breakpoint at PR. We found no significant correlations between baseline gut bacteria (alpha diversity before fentanyl IVSA) and mean infusions at FR1 or mean breakpoint at PR (Appendix A), suggesting that gut bacteria do not influence the motivation to self-administer fentanyl infusions.

### 3.5. Gut Bacteria Alpha Diversity Predicts Progressive Ratio Responding in Females

In addition to baseline gut bacteria influencing self-administration behavior, we wanted to consider the relationship between the rewarding value of fentanyl and gut bacteria diversity. We analyzed the correlation between gut bacteria alpha diversity after fentanyl IVSA and the mean breakpoint at PR. In females only, we discovered a positive correlation between Shannon index values and mean breakpoint at 1.25 μg/kg/infusion (*p* = 0.02), and a negative correlation between these same variables at 2.5 μg/kg/infusion (*p* = 0.002). These data show that at 1.25 μg/kg/infusion, gut bacteria alpha diversity increases breakpoint, while at 2.5 μg/kg/infusion, gut bacteria alpha diversity decreases breakpoint (Figure 3D).

### 3.6. Fentanyl Self-Administration Does Not Alter Beta Diversity of Gut Bacteria 

Beta diversity values were ordinated for visual representation using non-metric multidimensional scaling (NMDS) in R Studio (Figure 4A), as well as extracted from a distance matrix into individual Bray–Curtis dissimilarity values by comparing the distance before and after fentanyl IVSA within each animal (Figure 4B). A 2-way ANOVA evaluating the roles of sex and fentanyl dose on Bray–Curtis dissimilarity did not yield any significant findings (F(5,30) = 0.54 *p* = 0.74).

### 3.7. Gut Bacteria Beta Diversity Predicts Progressive Ratio Responding in Females

To evaluate the relationship between shifts in gut microbiota composition and the rewarding value of fentanyl, we tested the correlation between gut bacteria beta diversity and mean breakpoint at PR. Bray–Curtis dissimilarity measures how different two microbiota compositions are from each other. We found a positive correlation in females at 2.5 μg/kg/infusion (*p* = 0.025) (Figure 4C), indicating higher motivation for fentanyl as the microbiome shifts.

## 4. Discussion

The present study provides a feasible animal model of intravenous fentanyl self-administration and a bidirectional relationship between fentanyl self-administration and gut bacteria. We show here that fentanyl reinforcement and motivation are dependent on fentanyl dose, but not sex. Further, we show that both sex and dose determine the impact of fentanyl IVSA on alpha diversity of gut bacteria, as fentanyl IVSA increases diversity in males at 1.25 μg/kg/infusion and decreases alpha diversity in females at 1.25 vs. 0 μg/kg/infusion. Supporting a bidirectional relationship between fentanyl IVSA and gut bacteria, we find that bacterial diversity sex- and dose-dependently predicts responding for fentanyl infusions at a progressive ratio schedule of reinforcement.

Previous studies report sex differences in opioid abuse and reinforcement in humans and animals [33,34,35,36,37,38]. Although our results do not determine whether sex affects fentanyl self-administration at FR or PR schedules of reinforcement, we uncover an interaction between gut bacteria and sex to impact fentanyl IVSA. The discrepancy in our results compared to prior findings may be explained by methodological differences, such as species and strain used, nose poke vs. lever presses, route of drug administration, self-administration vs. experimenter-administration, and duration of drug access or exposure.

We show sex differences in baseline alpha diversity of gut microbiota, which is consistent with our hypothesis. These microbial differences are driven by Verrucomicrobia, *Prevotella*, and *Akkermansia*. The available literature on the impact of sex on gut microbiota is inconsistent, with findings either supporting or rejecting sex differences in gut microbiota diversity and composition [39]. Genotype, strain, and species appear to be stronger determinants than the sex of the rodent microbiota [40,41]. However, sex is an important variable in gut microbiota and behavior in both clinical and preclinical studies [42,43]. Further research is necessary to establish the role of sex on the gut microbiome, though our findings demonstrate that sex impacts baseline gut bacteria and opioid-induced changes in bacteria.

Our results support the hypothesis that opioid use affects gut bacteria. We see increased alpha diversity following fentanyl IVSA in males that self-administer 1.25 µg/kg/injection, and a reduction in alpha diversity following fentanyl IVSA in females at 1.25 vs. 0 µg/kg/injection. In addition, we find correlations between the diversity of gut bacteria and fentanyl IVSA at PR in females, but not males. The increasing breakpoint with increasing microbiome changes, both in alpha and beta diversity, is more likely that gut bacteria influence self-administration rather than responses affecting gut bacteria, as bacterial shifts do not occur that quickly [44]. Future studies aim to examine how direct manipulation of gut bacteria influences fentanyl self-administration.

The increase in alpha diversity after fentanyl IVSA in males is surprising and does not mimic results found in prior work focused on chronic morphine treatment in mice [10,12,13,45]. There are major differences in methodology from the animals examined, both in species and model, (e.g., opioid-dependent, self-administering), and drugs used. Additionally, there are arguments in favor of microbial composition rather than diversity, as high diversity is not necessarily “healthier” [46]. One possibility of increased alpha diversity in our results is an introduction of pro-inflammatory microbes [47], although further analysis of species is essential to clarify this. We observed a difference in *Verrucomicrobia* (phylum), *Ruminococcus* (genus), and *Akkermansia* (genus) in males at 1.25 µg/kg/injection, and *in Prevotella* (genus) in females at 1.25 µg/kg/injection. The lack of change in the Firmicutes/Bacteroidetes ratio before and after fentanyl IVSA is expected, as these two phyla dominate a stable adult gut microbiome and are less susceptible to disruption [48] than other phyla.

The present study is limited by its relatively small sample size, which obscures the natural variability of the gut microbiome. Further, as a microbiome study, our use of a conventional environment over germ-free conditions introduces contamination as a potential factor in both natural gut bacterial shifts and in sample processing. Although previous studies have observed microbiota differences following short-term drug treatment, our 11-day paradigm may be too short to see noticeable microbial changes. Co-housing of animals also may mask potential microbial shifts. Finally, we would argue both as a limitation and a strength that fentanyl dose alone is not sufficient to reveal fentanyl-induced changes in gut microbiota, as animals self-administer varying amounts due to individual preference. Additional experimentation could examine how a fixed fentanyl intake impacts gut bacteria, although experimenter-administered vs. self-administered drug may be a stressor.

Collectively, our data show the impact of intravenous fentanyl self-administration on gut microbiota composition, as well as the role of gut microbiota on fentanyl self-administration, in wild-type Sprague Dawley rats in a sex- and dose-dependent manner. Given the associations between the gut microbiota and stress, mood, psychiatric disorders, and behavior, evaluating the role of the gut microbiota in fentanyl use is a unique approach that could lead to new paths for the treatment of addiction. By identifying how the gut–brain axis influences fentanyl use, our research has the potential to significantly progress our understanding of the mechanisms influencing opioid addiction. The next steps are to investigate possible mechanisms underlying opioid-induced changes in gut bacteria, such as inflammation driven by decreased gut permeability (gut “leakiness”) and/or functional changes in gut peptides or bacterial metabolites, (i.e., short-chain fatty acids). Identification of direct mechanisms allows for targeted pharmacological therapeutics to address opioid abuse. Plausible clinical approaches include (1) early diagnosis and prevention by recognizing biomarkers or (2) supplementation with beneficial bacterial strains or metabolites, targeted anti-inflammatories, or microbiota transplantation.

## Figures and Tables

**Figure 1 microorganisms-10-01127-f001:**
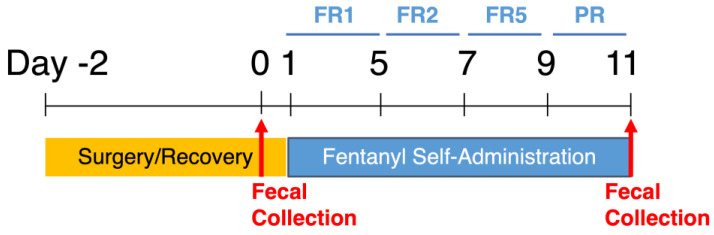
Experimental timeline for fentanyl self-administration and fecal sample collection. Animals undergo catheter implantation surgery and recover for 2 full days before starting self-administration. Animals self-administer fentanyl at fixed ratio (FR) 1 for 5 days, FR2 for 2 days, FR5 for 2 days, and progressive ratio (PR) for 2 days, for 11 days total. One fecal sample is collected from each animal the day before self-administration (day 0) and the last day of PR (day 11).

**Figure 2 microorganisms-10-01127-f002:**
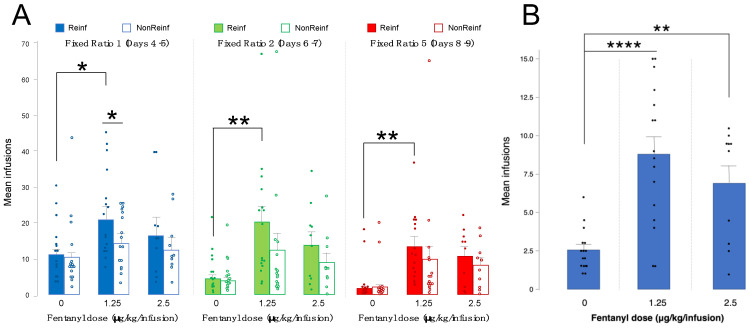
Fentanyl intravenous self-administration. (**A**) Mean number of infusions across 2 days at fixed ratio (FR) 1, FR2, and FR5 schedules of reinforcement, reinforced (Reinf, filled bars and circles) vs. non-reinforced (NonReinf, empty bars and circles) responses, at 0, 1.25, and 2.5 μg/kg/infusion (error bars represent S.E.M., ** *p* < 0.01, * *p* < 0.05, *n* = 10–16/group). (**B**) Mean infusion breakpoint over 2 days at a progressive ratio schedule of reinforcement at 0, 1.25, and 2.5 μg/kg/infusion (error bars represent S.E.M., **** *p* < 0.0001, ** *p* < 0.01, *n* = 10–16/group). Data are collapsed by sex.

**Figure 3 microorganisms-10-01127-f003:**
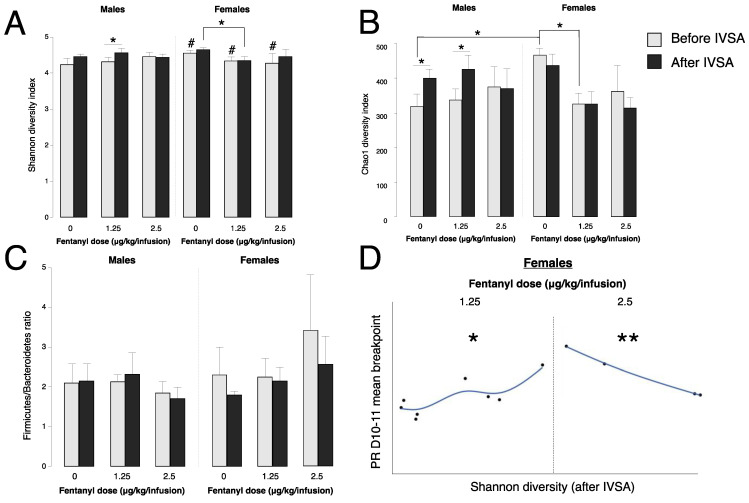
Alpha diversity of gut bacteria before and after IVSA. (**A**) Shannon diversity index values of gut bacteria before (gray bars) and after (black bars) IVSA in males and females (error bars represent S.E.M., * *p* < 0.05, # *p* < 0.05 vs. males, *n* = 4–9/group). (**B**) Chao1 diversity index values of gut bacteria before (gray bars) and after (black bars) in males and females (error bars represent S.E.M., * *p* < 0.05, *n* = 4–9/group). (**C**) Firmicutes/Bacteroidetes ratios before (gray bars) and after (black bars) IVSA in males and females (error bars represent S.E.M., *n* = 4–9/group). (**D**) Correlation plots of alpha diversity after IVSA vs. progressive ratio (PR) mean breakpoint on days (D) 10–11 in females at 1.25 and 2.5 μg/kg/infusion (** *p* < 0.01, * *p* < 0.05, *n* = 4–8/group).

**Figure 4 microorganisms-10-01127-f004:**
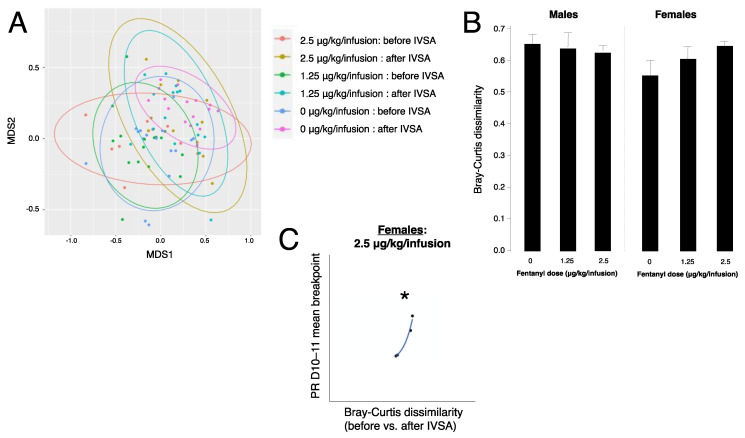
Beta diversity of gut bacteria before and after IVSA. (**A**) Non-metric multidimensional scaling (NMDS) ordination of gut microbiota communities before and after fentanyl IVSA at 0, 1.25, and 2.5 μg/kg/infusion. (**B**) Bray-Curtis dissimilarity values extracted from distance matrix comparing before and after IVSA within each animal at 0, 1.25, and 2.5 μg/kg/infusion in males and females (error bars represent S.E.M., *n* = 4–9/group). (**C**) Correlation plot of Bray–Curtis dissimilarity vs. progressive ratio (PR) mean breakpoint on days (D) 10–11 in females at 2.5 μg/kg/infusion (* *p* < 0.05, *n* = 4).

**Table 1 microorganisms-10-01127-t001:** Fentanyl intravenous self-administration. Mean number of infusions self-administered at fixed ratio (FR) 1, FR2, FR5, and progressive ratio (PR) schedules of reinforcement at 0, 1.25, and 2.5 μg/kg/infusion.

Fentanyl Dose	Number of Animals	Results
0 μg/kg/infusion	15	Mean infusions: FR1 D4–5 Reinf: 11.80 FR1 D4–5 NonReinf: 10.60 FR2 D6–7 Reinf: 6.26 FR2 D6–7 NonReinf: 5.76 FR5 D8–9 Reinf: 4.0 FR5 D8–9 NonReinf: 4.26 PR: 2.56
1.25 μg/kg/infusion	16	Mean infusions: FR1 D4–5 Reinf: 20.90 FR1 D4–5 NonReinf: 14.65 FR2 D6–7 Reinf: 20.43 FR2 D6–7 NonReinf: 12.53 FR5 D8–9 Reinf: 13.25 FR5 D8–9 NonReinf: 9.53 PR: 8.81
2.5 μg/kg/infusion	10	Mean infusions: FR1 D4–5 Reinf: 16.45 FR1 D4–5 NonReinf: 12.40 FR2 D6–7 Reinf: 13.80 FR2 D6–7 NonReinf: 8.85 FR5 D8–9 Reinf: 10.70 FR5 D8–9 NonReinf: 8.15 PR: 6.90

**Table 2 microorganisms-10-01127-t002:** Bacterial groups impacted by fentanyl intravenous self-administration (IVSA).

Bacterial Phylum or Genus	Results
*Verrucomicrobia*	Decreased after fentanyl IVSA (males, 1.25 μg/kg/infusion; *p* = 0.03) Increased in females vs. males before fentanyl IVSA (*p* = 0.01)
*Prevotella*	Increased after fentanyl IVSA (females, 1.25 μg/kg/infusion; *p* = 0.02) Decreased in females vs. males before fentanyl IVSA (*p* = 0.02)
*Ruminococcus*	Increased after fentanyl IVSA (males, 1.25 μg/kg/infusion; *p* = 0.03)
*Akkermansia*	Decreased after fentanyl IVSA (males, 1.25 μg/kg/infusion; *p* = 0.03) Increased in females vs. males before fentanyl IVSA (*p* = 0.01)

## Data Availability

Not applicable.

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
