# Peer review of "Dose- and Sex-Dependent Bidirectional Relationship between Intravenous Fentanyl Self-Administration and Gut Microbiota"

_microorganisms, 2022, doi:10.3390/microorganisms10061127_

Round 1

Reviewer 1 Report

Figure 2B. Bars seem to have been recolored in a second program and as a result have shifted in both the chart area and legend. Formatting in Figure 2B must be corrected prior to acceptance. Over-tracing in data representation is generally bad practice as it may lead to small changes in data representation (eg. bar height).

Line 201-202 'gut bacteria alpha diversity breakpoint decreases breakpoint.' -Omit first “breakpoint”

Manuscript would benefit from the following:

  • Discussion of which phyla were responsible for changes in diversity.
  • Resilience of Firmicutes/Bacteroidetes (aka core microbiome) should be mentioned in conclusions.

In future experiments, it is recommended to gather a larger pre-treatment dataset so that natural variability in the microbiome is better accounted for. 

Author Response

Point 1: Figure 2B. Bars seem to have been recolored in a second program and as a result have shifted in both the chart area and legend. Formatting in Figure 2B must be corrected prior to acceptance. Over-tracing in data representation is generally bad practice as it may lead to small changes in data representation (eg. bar height).
Response 1: Formatting has been fixed.

Point 2: Line 201-202 'gut bacteria alpha diversity breakpoint decreases breakpoint.' -Omit first “breakpoint”
Point 2: Thank you for bringing this to our attention.  This typo has been fixed.

Point 3: Manuscript would benefit from discussion of which phyla were responsible for changes in diversity.
Response 3: We have added this analysis and discussion in the results.

Point 4: Resilience of Firmicutes/Bacteroidetes (aka core microbiome) should be mentioned in conclusions.
Response 4: We have added this point in the discussion.

Point 5: In future experiments, it is recommended to gather a larger pre-treatment dataset so that natural variability in the microbiome is better accounted for. 
Response 5: We appreciate this insight and will take it into account in future experiments.  We also added this point in our discussion as a limitation.

Reviewer 2 Report

Review of manuscript with tittle:

Dose- and Sex-Dependent Bidirectional Relationship Between Intravenous Fentanyl Self-Administration and Gut Microbiota.

Authors: Michelle Ren and Shahrdad Lotfipour.

The paper presents an interesting topic. And the research itself is well done. I have reviewed the unrevised and already revised format. I have rectified the review report when I saw the revised version already reviewed by other reviewers. In general, I agree that the rectifications significantly improve the critical points that were initially shown. Nevertheless, my opinion is  minor revision with the following indications.

- Line 68: This is where I would include the distribution of rats, number of rats per cage and cages.

- How is the duration of 11 days determined? It is possible that the times set are too short to see noticeable changes in the microbiota.

- Line 86: explain reinforcement using the progressive ratio, it is not very clear.

- Line 88: The optimal doses of fentanyl administered (both delivered and booster) according to what criteria they were established. It is referenced, but I think it is convenient that this information is indicated in the manuscript.

- Line 94: How is the anesthetic permeability test involved in the trial? What is its purpose?

- Line 98: A recommendation for other occasions about the use of DNA/RNA Shield, is that it is not necessary to use it as long as the samples taken are immediately frozen at -80ºC. It is advisable to do everything under sterile conditions.

- Line 106: Indicate the reference of the use of the QUIME2 software.

- Figure 2: give the graphs the same size and make them as large as the format allows.

- Line 169: recommended to include the CHAO2 richness index and state the possible differences.

- Graph 3, improve resolution, match font, increase size and improve quality of bar graph.

- Line 180: changes have been seen at the level of relative abundance of the phylum Verrucomicrobia, but what happens at lower levels? Is there any genus that stands out as increasing or decreasing?

- Recommended to include a table with the doses, their distribution, and the results obtained. It is confusing and difficult to understand in the text.

- Lines 212,217 correspond to materials and methods.

- Discussion, if there are already differences in the basal diversity between males and females, it is evident that there will be differences in the final diversity, but is this unit of change similar in both after the different treatments?

- Which bacterial groups show the greatest difference in relative abundance between males and females? This is a part from which more information can be extracted and more concrete results can be shown.

- If the diversity indexes increase, do the richness indexes also increase?  Which genera contribute to this increase? Are there precedents for any of the genera in opioid treatments? Have you looked for the possible presence of biomarkers that promote this increase in diversity?

- Line 283, I think the cohabitation of the animals in the same cage contributes to the uniformity of the results in the microbiota, it would be more similar between them.

Author Response

Dear Microorganisms,

We appreciate the opportunity to revise our manuscript. We believe the reviewers’ constructive recommendations have substantially improved our manuscript. Attached are our responses to their comments. All new changes have been tracked using “Track Changes” in Microsoft Word. We would like to thank the reviewers and editors for their time.

On behalf of all authors:
Sincerely,
Michelle Ren

--

Point 1: Line 68: This is where I would include the distribution of rats, number of rats per cage and cages.
Response 1: We have now included this information where recommendation (in the “Animals” section of Methods).

Point 2: How is the duration of 11 days determined? It is possible that the times set are too short to see noticeable changes in the microbiota.
Response 2: The duration of self-administration is based on prior work (both in and out of our lab) establishing self-administration at the schedules of reinforcement used in this study. We have now added in the discussion the potential limitation of this duration of time in observing microbiota changes.

Point 3: Line 86: explain reinforcement using the progressive ratio, it is not very clear.
Response 3: We have now added further explanation of the progressive ratio schedule of reinforcement where indicated.

Point 4: Line 88: The optimal doses of fentanyl administered (both delivered and booster) according to what criteria they were established. It is referenced, but I think it is convenient that this information is indicated in the manuscript.
Response 4: We have added the rationale of our selected fentanyl doses based on prior studies’ criteria.

Point 5: Line 94: How is the anesthetic permeability test involved in the trial? What is its purpose?
Response 5: As the catheters administer drug intravenously, the anesthetic (propofol) is used to test patency of the catheters. Propofol acts rapidly and is short-lasting, so evidence of anesthesia following propofol administration indicates the catheter is still patent and the drug self-administered represents the drug the animal is receiving.

Point 6: Line 98: A recommendation for other occasions about the use of DNA/RNA Shield, is that it is not necessary to use it as long as the samples taken are immediately frozen at -80oC. It is advisable to do everything under sterile conditions.
Response 6: We appreciate this recommendation and will take this into consideration for our future studies.

Point 7: Line 106: Indicate the reference of the use of the QUIME2 software.
Response 7: We have updated this to reference use of QIIME2.

Point 8: Figure 2: give the graphs the same size and make them as large as the format allows.
Response 8: We have edited the formatting to fit these recommendations.

Point 9: Line 169: recommended to include the CHAO2 richness index and state the possible differences.
Response 9: We have now added Chao1 richness data (Chao2 was not available in our software).

Point 10: Graph 3, improve resolution, match font, increase size and improve quality of bar graph.
Response 10: We have now edited the formatting of Figure 3, as well as added a new graph.

Point 11: Line 180: changes have been seen at the level of relative abundance of the phylum Verrucomicrobia, but what happens at lower levels? Is there any genus that stands out as increasing or decreasing?
Response 11: We completed a further analysis to observe lower levels and found either a reduction or increase in relative abundance in specific genera. We have added these data into a table (Table 2).

Point 12: Recommended to include a table with the doses, their distribution, and the results obtained. It is confusing and difficult to understand in the text.
Response 12: We added a table to represent these data (Table 1).

Point 13: Lines 212,217 correspond to materials and methods.
Response 13: We have removed line 212 and rewritten line 217 so it fits better with the presentation of the results.

Point 14: Discussion, if there are already differences in the basal diversity between males and females, it is evident that there will be differences in the final diversity, but is this unit of change similar in both after the different treatments?
Response 14: Because the basal differences between males and females are that males have decreased diversity, and fentanyl self-administration increases diversity in males, the difference between males and females after self-administration essentially disappears.

Point 15: Which bacterial groups show the greatest difference in relative abundance between males and females? This is a part from which more information can be extracted and more concrete results can be shown.
Response 15: We have added data showing sex differences in bacterial relative abundance in Table 2.

Point 16: If the diversity indexes increase, do the richness indexes also increase? Which genera contribute to this increase? Are there precedents for any of the genera in opioid treatments? Have you looked for the possible presence of biomarkers that promote this increase in diversity?
Response 16: Since Shannon diversity measures both richness and evenness, while Chao1 diversity estimates richness based on relative abundances, the richness index may increase with Shannon diversity index. However, there is not a direct relationship between the two indices. This is shown in our analyses of both Shannon and Chao1, as the statistics do not match in both diversity measures. From our understanding, there has not been a focus on specific genera in opioid treatment yet, but prior studies have identified genus and species differences from chronic opioid use. We have not examined possible biomarkers that promote increased diversity, which is a valuable idea and added to the future directions in our discussion.

Point 17: Line 283, I think the cohabitation of the animals in the same cage contributes to the uniformity of the results in the microbiota, it would be more similar between them.
Response 17: Thank you. We have rewritten this sentence for clarification and accuracy.

Reviewer 3 Report

The authors present a novel experimental rodent study on the two-way relationship between the gut microbiome and opioid self-administration. They report several interesting relationships, the implications of which remain to be determined. This is a well-designed study with a well written manuscript. This manuscript might be improved by considering the following comments.

Abstract: More details on experimental design are warranted. For example, how many rats were used in this study?

Abstract: The presentation of the results could be more precise. For example, “Alpha diversity is increased or decreased following fentanyl…” is not very clear. Ideally, some numerical data would be presented, including some quantification of the magnitude and statistical significance of the findings.

Methods: As far as I can tell, there is no information on the number of rats in each group. This is essential information.

Please define the acronym “PR” the first time it is used.

Results: The experimental timeline seems like it belongs in the methods section.

3.4: Sentence 3: “We observed an overall between…” There seems to be several typos in this sentence that make it difficult to understand. Please consider rewording.

Figure 2A: The absolute differences in alpha diversity between groups is quite small. It seems that the statistical significance is mainly driven by very low standard deviations. I would like more information on the number of rats in each group at the start of the experiment, the amount of data that was discarded for technical issues (iv access failure, rat death, etc), and the amount of data that was discarded because of extreme variation from the mean. Regardless of these details, these small differences should not be overemphasized in the discussion section on the implications of these findings.

Discussion: Please include a limitations paragraph

Please elaborate on the translatability of this study and future similar studies. A decade from now, how could this line of research impact clinical medicine? Do the authors foresee a pharmacologic treatment or perhaps a diagnostic/prognostic test stemming from this? If so, please elaborate on the next research steps. What studies are needed to better understand these relationships between the gut microbiome and opioids that could facilitate the clinical translation of this research?

Author Response

Point 1: Abstract: More details on experimental design are warranted. For example, how many rats were used in this study?
Response 1: We have now added this information in the abstract as well as the methods.

Point 2: Abstract: The presentation of the results could be more precise. For example, “Alpha diversity is increased or decreased following fentanyl…” is not very clear. Ideally, some numerical data would be presented, including some quantification of the magnitude and statistical significance of the findings.
Response 2: We added in the abstract that alpha diversity was measured using the Shannon diversity index, which takes into account both richness and evenness of gut bacteria.  We have also rewritten this sentence so it is more clear.

Point 3: Methods: As far as I can tell, there is no information on the number of rats in each group. This is essential information.
Response 3: We have now added animal numbers used and the groups they were randomly assigned to.

Point 4: Please define the acronym “PR” the first time it is used.
Response 4: Thank you, now addressed.

Point 5: Results: The experimental timeline seems like it belongs in the methods section.
Response 5: We have moved the experimental timeline from the results to the methods.

Point 6: 3.4: Sentence 3: “We observed an overall between…” There seems to be several typos in this sentence that make it difficult to understand. Please consider rewording.
Response 6: We have reworded this sentence for better clarity.

Point 7: Figure 2A: The absolute differences in alpha diversity between groups is quite small. It seems that the statistical significance is mainly driven by very low standard deviations. I would like more information on the number of rats in each group at the start of the experiment, the amount of data that was discarded for technical issues (iv access failure, rat death, etc), and the amount of data that was discarded because of extreme variation from the mean. Regardless of these details, these small differences should not be overemphasized in the discussion section on the implications of these findings.
Response 7: We have updated the methods so that the rat numbers are included. We also expanded on findings at the phylum level to support that the alpha diversity differences are subtle.

Point 8: Discussion: Please include a limitations paragraph.
Response 8: We have added limitations to our discussion section.

Point 9: Please elaborate on the translatability of this study and future similar studies. A decade from now, how could this line of research impact clinical medicine? Do the authors foresee a pharmacologic treatment or perhaps a diagnostic/prognostic test stemming from this? If so, please elaborate on the next research steps. What studies are needed to better understand these relationships between the gut microbiome and opioids that could facilitate the clinical translation of this research?
Response 9:
We now added in our discussion future directions and clinical translatability of this study.

Round 2

Reviewer 3 Report

The authors have provided satisfactory responses to my comments and questions and made the corresponding changes to the manuscript. I hope they will agree that this version of the manuscript is stronger than the original.

Author Response

We would like to thank the reviewers and editors for their time.